# The Interaction of Aeolian Sand and Slope on Runoff and Soil Loss on a Loess Slope via Simulated Rainfall under Laboratory Conditions

**Zongping Ren [1,2], Jinjin Pan [1], Zhanbin Li [1], Peiqing Xiao [2,*], Zhenzhou Shen [2], Lu Jia [1] and Xiaozheng Li [1]**

[1]  State Key Laboratory of Eco-Hydraulics in Northwest Arid Region of China, Xi'an University of Technology, Xi'an 710048, China
[2]  Key Laboratory of Soil and Water Conservation on the Loess Plateau of Ministry of Water Resources, Yellow River Institute of Hydraulic Research, Zhengzhou 450003, China
*  Correspondence: peiqingxiao@163.com

**Abstract:** The wind–water erosion crisscross region, where the topography is complicated, is the most severe area of soil erosion on the Loess Plateau. The wind and terrain both have an impact on the soil water erosion process. In order to evaluate the effects of sand cover on runoff and soil loss characteristics, a series of experiments was conducted in two contrasting treatments. One treatment was a bare loess soil slope serving as the control, and the others were sand-covered loess slopes with five different slopes. The results showed that the runoff time, total runoff yield, and total soil loss were different between the sand-covered slope and the loess slope on the slope of 15°. The sediment concentration of the sand-covered slope was significantly higher than that of the loess slope during the entire rainfall process ($p < 0.05$). The increase in the slope gradient shortened the surface runoff initiation times and enhanced the total runoff volume and soil loss. The total runoff volume and the total soil loss were 39.7 L and 44.3 kg, respectively, on the sand-covered slope of 10°. When the slope gradient increased from 10° to 30°, the total runoff volume and the total soil loss increased by 22.8 L and 42.8 kg, respectively, while the surface runoff initiation times shortened by 300 s. For the sand-covered slopes, the erosion processes appeared to be mainly dominated by sediment transport. The correlation between soil loss rates and slope gradients demonstrated the secondary polynomial function. In addition, the critical slope of sand-covered slopes was from approximately 23° to 28°. The proportion of sand cover and slope responsible for soil erosion was 3:1, which means the wind effect was more important than the terrace factor in terms of soil water erosion in the wind–water erosion crisscross region. The results provide a theoretical basis for soil erosion control in this area.

**Keywords:** sand cover treatment; slope gradient; critical slope; simulated rainfall

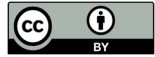

## 1. Introduction

The wind–water erosion crisscross region has experienced the most serious soil erosion on the Loess Plateau, which is mainly located in the region adjacent to the provinces of Shanxi, Shaanxi, and Inner Mongolia. This region suffers from severe combined wind and water erosion on a temporal scale [1–3]. Strong winds drive aeolian sand materials to rills, gullies, and river channels in the winter and spring, providing sediment for further erosion from summer rains and runoff [4]. There are different thicknesses of sand deposited on loess slopes, which has formed sand-covered loess slopes.

The sand-covered loess slopes are obviously different from the non-sand-covered loess slopes, as is mainly reflected in the particle size composition, soil aggregates, soil infiltration, and soil erodibility [5]. These characteristics could affect runoff and sediment yield. Many studies have also shown that after wind erosion, sand cover could

significantly delay the runoff time, change the runoff generation mechanism, and increase the erosion amount by several fold [4–6]. Sand cover has been shown to be the main factor related to increased erosion. The slope also affects the degree of erosion. Zhang showed that the characteristics of slope erosion also increased with the change in slope gradient under the same thickness of sand cover [7]. However, a few studies have focused on quantifying the interaction effect of wind erosion and slope.

The slope gradient, which is one of the prominent factors affecting land surface-hydrological processes such as runoff and infiltration, has significant impacts on soil erosion processes [8–13]. Many studies showed that soil erosion begins to reach a critical value when the slope reaches 25° to 35°, and the critical slope changes due to differences in topography and landform [14]. The critical slope for erosion varies in different areas of the Loess Plateau, which is mainly between 22° and 38°, with erosion gradually decreasing after reaching its peak [11]. A study on slope erosion in the Yimeng mountainous region showed that soil erosion increased significantly with increasing slope, reaching a maximum at 25° [15]. Li et al. also drew a similar conclusion by analyzing the characteristics of runoff and sediment production on black soil slopes [16]. It can be seen that the effects of slope on the erosion process vary in different regions, and the value of the critical slope is also different. Some studies have also found that the critical slope is related to rainfall characteristics and soil texture [14].

As mentioned above, sand cover changes the soil characteristics and significantly affects the runoff amount and the sediment yield in the wind–water erosion crisscross region [4–7]. Therefore, whether the relationship between slope and erosion changes when eroded material changes, and which factor plays a dominant role in soil water erosion between the slope and wind erosion effect remains unclear. Therefore, sand-covered slopes of different slope gradients were used to simulate the effects of slope and wind erosion. The result will be useful to understand the interaction effect of the slope gradient and wind-blown sand cover on water erosion in the wind–water erosion crisscross region.

## 2. Materials and Methods

### 2.1. Materials

The simulated rainfall laboratory experiment was carried out in the State Key Laboratory of Eco-hydraulics in the Northwest Arid Region of China in Xi'an, China. The soil used in this experiment was from the Liudaogou watershed in Suide County, Shaanxi Province, and the aeolian sand was from a dune located near the tested soil. The soil was composed of 24.8% sand, 69.4% silt, and 5.8% clay, and the sand was composed of 97.5% sand, 2.0% silt, and 0.5% clay. The loess soil is sandy loam, and the aeolian sandy soil is considered sandy soil by the soil classification standard of the United States Ministry of Agriculture (USDA).

A soil flume was used for rainfall simulation, which was 2 m in length, 0.75 m in width, and 0.40 m in height. Before the experiment, a 5 cm thick aeolian sand layer was laid on the bottom of the flume to ensure the water permeability of the experimental soil, which was close to the natural state. The soil layer above the bottom layer was 25 cm thick. The loess soil bulk density was controlled at about 1.40 g/cm³. In addition, for the sand cover treatment, the top layer was a 2 cm layer of aeolian sand that was uniformly spread over the loess soil surface. The initial soil moisture content was controlled at about 15%.

A self-designed upward spraying rainfall simulator was used in this experiment. The rainfall intensity was set for 1.5 mm min$^{-1}$ with distribution uniformity of over 85% of the area. In addition, the rainfall duration was one hour (Figure 1). There were six treatments in the experiment; two replicates were made for each treatment. Conditions of the rainfall and the dimensions of the soil slope are shown in Table 1.

**Table 1.** Conditions of the rainfall and the dimensions of soil slope.

| Treatment | Slope Angles (°) | Treatment Mode | Rainfall (mm min⁻¹) |
|---|---|---|---|
| Loess slope (control) | 15 | Two layers: top layer was 25 cm loess slope; the bottom layer was 5 cm aeolian sand | 1.5 |
| Sand cover treatment 1 | 10 | | 1.5 |
| Sand cover treatment 2 | 15 | Three layers: top layer was a 2 cm layer of aeolian sand; the | 1.5 |
| Sand cover treatment 3 | 20 | middle layer was 25 cm loess slope; the bottom layer was | 1.5 |
| Sand cover treatment 4 | 25 | also 5 cm sand. | 1.5 |
| Sand cover treatment 5 | 30 | | 1.5 |

Runoff samples were collected in buckets every minute after runoff occurred. The samples were weighed and then oven-dried at 105 °C to calculate sediment yields and runoff volumes.

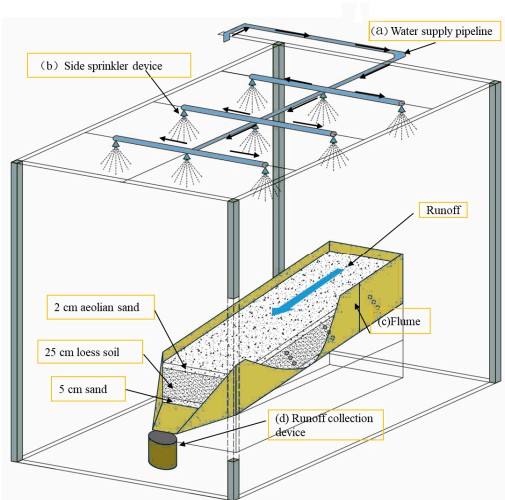

**Figure 1.** Experimental devices including water supply pipeline (**a**), sprinkler device (**b**), flume (**c**), and runoff collection device (**d**).

*2.2. Data Processing*

In this paper, variance decomposition analysis was used to calculate the effect of sand cover thickness and slope on erosion and sediment yield. The variance decomposition analysis provides a multivariate framework where changes in a particular variable are related to the autoregressive process of all dependent variables as well as contemporaneous values of all exogenous variables. Variance decomposition functions demonstrate how each dependent variable contributes to changes in the dependent variable. In this study, slope and sediment thickness were taken as the independent variables, and erosion rate was taken as the dependent variable. The model was specified as follows [17]:

$$RVC_{j \to i}(\infty) = \frac{\sum_{q=0}^{\infty} (c_{ij}^{(q)})^2 \sigma_{ij}}{\text{var}(y_i)} = \frac{\sum_{q=0}^{\infty} (c_{ij}^{(q)})^2 \sigma_{ij}}{\sum_{j=1}^{k} \left\{ \sum_{q=0}^{\infty} (c_{ij}^{(q)})^2 \sigma_{ij} \right\}}, i, j = 1, 2, ..., k \tag{1}$$

*RVC* is the variance contribution rate, in which *k* is the number of variance decomposition terms of erosion rate (*y*); *c* is the coefficient; σ is the standard deviation; and variance (*$y_i$*) is the sum of variance decomposition terms of the erosion rate. There were 40 samples in

total, including the 12 samples in this paper and 28 samples in the paper of Zhang et al., which were used to analyze the impact of sediment thickness and slope on erosion and sediment yield[7].

The coefficient of variation (*CV*) was used to analyze the dispersion degree of runoff and sediment yield per minute, and the calculation formula is given as follows:

$$CV = \sigma/\mu \tag{2}$$

where $\sigma$ is the standard deviation and $\mu$ is the mean value. All of the above statistical analyses including a *t*-test were performed using R3.4.

### 3. Results and Discussion

*3.1. Runoff*

Table 1 shows the differences in the runoff initiation times and total runoff volume between the sand-covered loess slopes and the loess slope. On the 15° slope, the runoff initiation times for the sand-covered loess slope were 8.1 times greater than those of the loess slope (Table 2); the total runoff volume of the loess slope was 2.3 times greater than that of the sand-covered loess slope. Meanwhile, the runoff processes could obviously be divided into two stages: the rapid growth stage and relatively stable stage (Figure 2). The runoff rates on the loess slope increased quickly within 4–18 min and then remained in a relatively stable stage. The runoff rates of the sand cover treatments increased sharply approximately 15 min after runoff initiation and then decreased slowly with fluctuations in the relatively stable stage (Figure 3). The coefficient of variation (CV) of the runoff rates of the loess slope was 0.20, while the CV of the runoff rates of the sand cover treatments was more than 0.40.

For the sand cover treatment slopes, the runoff initiation times of the 10° slope was 1393 s. When the sand-covered loess slopes increased from 10° to 30°, runoff initiation times decreased by 301 s (Table 2). The total runoff volume increased with the slope increase; its values increased by 22.8 L. The results showed that the runoff process of the sand cover treatments demonstrated more fluctuations than that of the loess slope (Figures 1 and 2). The CVs of the runoff rates on 10°, 15°, 20°, 25°, and 30° sand-covered loess slopes were 0.40, 0.46, 0.65, 0.57, and 0.44, respectively.

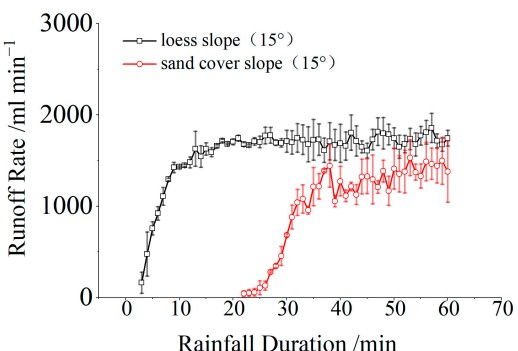

**Figure 2.** Runoff rates on the loess slopes with and without aeolian sand cover.

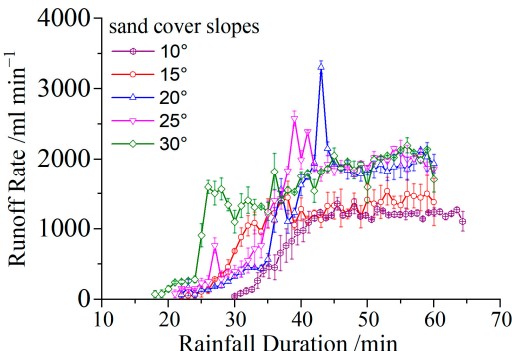

**Figure 3.** Change in erosion amount on sand-covered loess slope with different slopes.

Overall, more surface runoff was produced from the loess slope. We anticipated this response since a surface seal forms for the loess slope due to the raindrop impact. Once this seal forms, the surface runoff rate can be several orders higher than that of the unsealed soil [18,19]. This further supports the idea that the surface seal is favorable for runoff generation [18–20]. For the sand-covered loess slope, the raindrop energy dissipation by the sand layer was due to the larger porosity of the soil. The conditions are not favorable for surface seal formation. The dominant infiltration–excess runoff generation shifted to a prevailing saturation–excess runoff generation in the sand cover treatment. During the early stage of rainfall, most of the rainfall infiltrates into the slope, which makes the soil water content under the sand cover gradually reach a nearly saturated state, and then the slope begins to produce runoff. Previous research has also found that sand cover increased the soil infiltration and changed the runoff mode [4–7].

The increase in slope gradient shortened the surface runoff initiation times and enhanced the total runoff volume. In addition, the results of this study showed that there was a positive correlation between soil loss and slope gradient on sand-covered slopes, which was consistent with previous findings [7]. Such results are attributed to the fact that the slope gradient determines the magnitude of the downslope component force of gravity, which directly affects sand and soil movement during the rainfall process. During runoff, the runoff processes of the sand cover treatment demonstrated more fluctuations than the loess slope. The process resulted in extreme runoff rates on the 20°, 25°, and 30° slopes. This phenomenon can be explained by the stored water in the sand layer and soil layer; when it reached its maximum soil water-holding capacity, it was released instantaneously due to gravity and the erosion effect.

**Table 2.** The values of runoff and soil erosion in 60 min of simulated rainfall.

| Treatment | Slope/° | The Runoff Initiation Times/s | Total Runoff Volume/L | Total Sediment Yield/kg | First Collapse Time/min |
|---|---|---|---|---|---|
| Loess slope | 15 | 172 | 92.4 | 39.2 | No collapse |
| Sand-covered loess slope | 10 | 1393 | 39.7 | 44.3 | No collapse |
| | 15 | 1399 | 40.5 | 59.4 | No collapse |
| | 20 | 1339 | 49.8 | 83.1 | 43 |
| | 25 | 1231 | 54.9 | 84.5 | 40 |
| | 30 | 1091 | 62.5 | 87.1 | 36 |

*3.2. Soil Loss*

The sand cover treatment enhanced the soil loss. As is illustrated in Table 2, the total sediment yield was 59.4 kg under the sand cover treatment, which was almost 1.5 times higher than that of loess slope at the same slope gradient treatment. The total sediment yield increased with the increase in slope gradient, and the values increased by 42.8 kg when the slope gradient increased from 10° to 30°. The change trend in soil rates was similar to that of the runoff yield for sand-covered slopes. The soil loss process in Figure 4 showed more fluctuation than the runoff process. The CV values of the soil rates on the 10°, 15°, 20°, 25°, and 30° slopes were 0.43, 0.47, 0.74, 0.56, and 0.35, respectively.

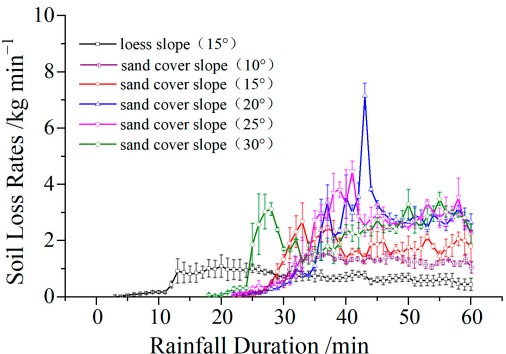

**Figure 4.** Soil loss rates of different slopes.

The increased total sediment yield will be further discussed below. Most of the rainfall in the early stage of the experiment was lost through infiltration, so there was no erosion in the early stage. When the water content of the sandy soil reaches the maximum water-holding capacity, the sandy soil begins to discharge. The cohesive force between soil particles of the saturated soil decreases, then the loosened soil particles are easier to detach by water flow [21]. The soil water content of the slope is nearly saturated under the treatment of covering sand, the hydrologic condition of the slope is favorable for the development of erosion [22,23]. It has been demonstrated that when the soil water content was close to saturation, the amount of soil erosion increased by approximately 24% [4]. A paired t-test showed that the sediment concentration of the sand-covered slope was significantly higher than that of the loess slope during the whole rainfall process (Figure 5). Additionally, the sediment concentration of the sand-covered slope experienced little change when it began to generate runoff. Due to the weakened soil anti-erodibility and abundance of soil, the soil erosion process of sand-covered slopes was limited by the sediment transport capacity [24,25].

As the soil erosion process did not reach a stable state, which mainly involved rill sidewall erosion in the later stage. The sidewall slumping soil accumulated at the outlet and blocked the outlet. When the rainfall continued, the erosive energy continuously accumulated and reached the maximum value, then an extreme maximum value of flow yield and sediment yield rate appeared [12]. As a result of the abundant soil, the relationships between water and sediment in five slope gradient treatments were very consistent (Figure 6). Thus, it was further demonstrated that the erosion process is mainly affected by the sediment transport capacity on sand-covered slopes.

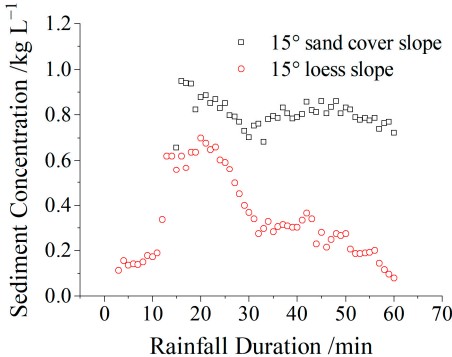

**Figure 5.** Sediment concentration of sand-covered and loess slopes (15°).

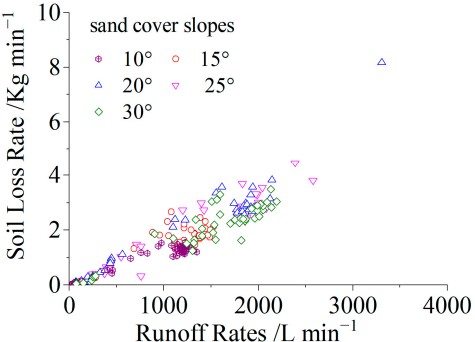

**Figure 6.** The relationship between runoff rate and soil loss rate on sand-covered slopes under different slope gradient.

### 3.3. Analysis of the Interaction between Slope and Sand Cover on Soil Loss

The soil erosion intensity was different between the sand cover treatment and loess slope. As is shown in Figure 7, the correlation between soil loss rates and slope gradient demonstrates the secondary polynomial function. For 1.5 cm of the sand-covered loess slope, the critical slope gradient was 28°. For 5 cm of the sand-covered loess slope, the peak value was 25°. Regarding the soil loss rates at 10 cm of the sand-covered loess slope, the critical slope gradient was 23°. The soil loss rate of the loess slope reached the threshold value at approximately 35°. In addition, the critical slope of the sand-covered slope was from approximately 23° to 28°, which is consistent with previous studies [12–16,24]. Figure 8 shows the effect of the slope gradient and sand layer thickness on soil erosion intensity. Those two factors explained 68% of the soil erosion intensities, in which the effect of the sand layer thickness factor on soil erosion intensity accounted for 75.0%, while the effect of the slope factor on the soil erosion intensity accounted for 25.0% (Figure 8). It can be seen that the effects of sand cover on soil erosion were higher than that of the slope, which demonstrated that wind erosion was the dominant factor. Xu also illustrated that erosion and the sediment yield of the wind–water erosion crisscross region was higher than other loess areas based on hydrometric station data, which indirectly provided evidence that the combined effect of aeolian sand and water was more significant than the terrain factors [26–28]. Our research further showed that aeolian sand processes were the dominant factor for soil water erosion in the wind–water erosion crisscross region. Of course, it should be mentioned that the findings should be further tested by field experience.

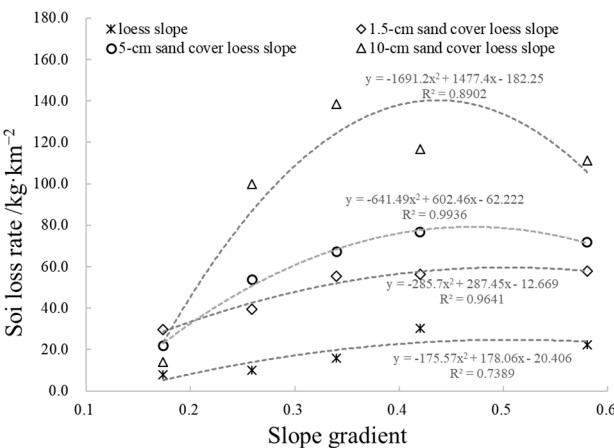

**Figure 7.** Soil loss rate under different slopes and sand cover thicknesses.

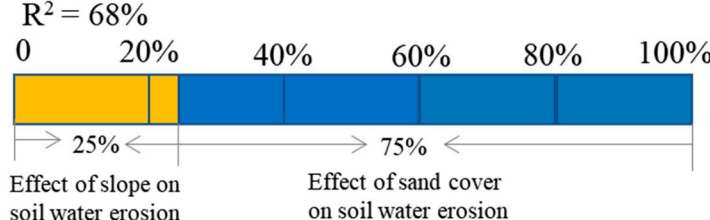

**Figure 8.** Percentage of effects of sand cover and slope on soil water erosion. ($R^2$ means 68% variance of the soil water erosion is explained by the effect of slope and sand cover, with about 75% of it belonging to sand cover and 25% of it belonging to slope.).

## 4. Conclusions

(1) The aeolian sand layer increased infiltration and delayed the runoff initiation time. The runoff time and total runoff yield were different between the sand-covered slope and the loess slope. The increase in the slope gradient shortened the surface runoff initiation times and enhanced the total runoff volume. The total runoff volume increased by 22.8 L, and the surface runoff initiation times shortened by 300 s when the slope gradient increased from 10° to 30°.

(2) Sand cover treatment enhanced the soil loss. The total soil loss of the sand cover treatment was 1.5 times higher than that of the loess slope. The sediment concentration of the sand-covered slope was significantly higher than that of the loess slope during the entire rainfall process ($p < 0.05$). With an increase in the slope gradient, the total sediment yield increased from 44.3 kg to 87.1 kg. For the sand-covered slopes, the erosion processes were mainly limited by sediment transport.

(3) The correlation between soil loss rates and slope gradients demonstrated the secondary polynomial function. The soil loss rate of the loess slopes reached the threshold value at approximately 35°. The critical slope of the sand-covered slope was from approximately 23° to 28°. The proportion of sand cover and slope affecting soil erosion was 3:1, which means the wind effect is more important than the slope factor for soil water erosion in the wind–water erosion crisscross region.

**Author Contributions:** Conceptualization, Z.R. and Z.L.; methodology, Z.L.; software, L.J.; formal analysis, J.P. and X.L.; investigation, X.L. and Z.S.; writing—original draft preparation, Z.R.; writing—review and editing, Z.R. and Z.L.; supervision, P.X.; project administration, P.X. and Z.S.; funding acquisition, P.X. All authors have read and agreed to the published version of the manuscript."

**Funding:** This work was supported by the open foundation of Key Laboratory of Soil and Water Conservation on the Loess Plateau of Ministry of Water Resources (201805); National Natural Science Foundation of China (52022081, 41601291).

**Data Availability Statement:** The data generated and/or analyzed during the current study are not publicly available.

**Conflicts of Interest:** The authors declare no conflict of interest.

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
