# Peer review of "The Interaction of Aeolian Sand and Slope on Runoff and Soil Loss on a Loess Slope via Simulated Rainfall under Laboratory Conditions"

_water, doi:10.3390/w15050888_

Round 1

Reviewer 1 Report

Review Comments

Extensive English edits are necessary as many expressions are awkward and rough.

Authors need to check and improve writing throughout the paper. Some problems should be clarified before the publication. I would suggest a major revision that includes the English edits and reorganize Introduction, and Results and discussions.

General comments:

1. There are many grammars and some typewriting problems. Please check the text carefully.

2. The introduction should be rewritten. Expressions and wordings are awkward. The authors should clarify the scientific problem clearly by citing more related references.

3. The authors demonstrated that a positive correlation was found between soil losses and slope gradient in sand cover slope. Could you please provide detailed information?

4. The authors indicated that 75% was explained by sand layer thickness factor, while 25% by slope. How to calculate these contributions? Please provide your detailed information to support these results.

Specific comments:

1. Section Introduction, I think “In addition to” is not suitable here.

2. “slope gradient is one of the most factors affecting soil erosion the processes of slopes” I cannot understand this sentence. Please rewrite.

3. “mainly between 22° and 38°”, What does it mean?

4. “reached a similar conclusion”, I think you should use draw.

5. “Therefore, sand-covered slope was … in wind-water erosion crisscross area”, What does this sentence mean? Please rewrite this sentence clearly.

6. Section 1.1 The soil was composed of 24.8%, … What du you mean by 24.8%? Please give details.

7. Section 2.1 “and then decreased slowly fluctuations in the relatively stable stage”, What does fluctuations mean in this sentence?

8. Section 2.1 “This phenomenon can be explained by the stored water … .. erosion effect”, Please rewrite this sentence.

9. “The process appeared extreme runoff rates on the 20, 25 and 30 slopes”, what do you mean by this sentence? Please rewrite this sentence.

10. “And the total sediment yield of and …  hand a significant from 20”, the authors should clarify had a significant what?

11. I cannot understand the meaning of the Figure 8. Please provide more meaningful figure.

Author Response

    First of all, the authors would like to thank the editor for the handling of our manuscript. We also sincerely thank to two anonymous reviewers for the review. The authors have carefully considered all issues mentioned in the reviewers' comments.

Sincerely Yours,

Dr. Zongping Ren et al.

Response to the Comments of Reviewer #1

1.There are many grammars and some typewriting problems. Please check the text carefully.

Response: Thanks for your comment. This paper has undergone English language editing by MDPI.

  1. The introduction should be rewritten. Expressions and wordings are awkward. The authors should clarify the scientific problem clearly by citing more related references.

   Response: Thanks for your comment. We rewritten the introduction, and also clarify the scientific problem. Under the interaction of wind erosion and slope, whether the relationship between slope and erosion changes when eroded material changes, and which factor plays a dominant role in the soil water erosion between the slope and wind erosion effect remains unclear. We add the sentence in the introduction.

3.The authors demonstrated that a positive correlation was found between soil losses and slope gradient in sand cover slope. Could you please provide detailed information?

Response: The total sediment yield increased with the increase in slope gradient, and the values increased by 42.8 kg when the slope gradient increased from 10° to 30°. (Table 2).

Table 2. The values of runoff and soil erosion in 60 minutes of simulated rainfall.

Treatment

Slope/°

The runoff initiation times/s

Total runoff volume/L

Total sediment yield/kg

First collapse time/min

Loess slope

15

172a

92.4a

39.2a

No collapse

Sand-covered loess slope

10

1393b

39.7b

44.3ab

No collapse

15

1399b

40.5b

59.4b

No collapse

20

1339b

49.8bc

83.1c

43

25

1231bc

54.9c

84.5c

40

30

1091c

62.5c

87.1c

36

There were significant differences among treatments using the ANOVA test (P<0.05).

  1. The authors indicated that 75% was explained by sand layer thickness factor, while 25% by slope. How to calculate these contributions? Please provide your detailed information to support these results.

Response: Thanks for your comment. We used the 40 total samples in total, including the 12 samples in this paper and 28 samples in the paper of Zhang et al, which were used to analysis of the impact of sediment thickness and slope on erosion and sediment yield. The variance decomposition analysis could demonstrate the dependent variable (such as rainfall, slope, sand cover thickness) contributes to change of the total erosion and sediment yield. The calculated result showed 68% variance of the soil water erosion is explained by the effect of slope and sand cover, with about 75% of it belonging to sand cover and 25% of it belonging to slope.

  1. Section Introduction, I think “In addition to” is not suitable here.

Response: Thanks for your comment. We rewrote this sentence as follows: “However, the slope also affects the degree of erosion.”

  1. “slope gradient is one of the most factors affecting soil erosion” I cannot understand this sentence. Please rewrite. 

Response: We rewrote this sentence as follows:“slope gradient is one of the most prominent factors affecting the soil erosion processes of slopes”

  1. “mainly between 22° and 38°”, What does it mean?

Response: Thanks. The critical slope for erosion of the Loess Plateau is not an invariant value.it varies in different areas, but the values is mainly 22° and 38°.

  1. “reached a similar conclusion”, I think you should use draw.

Response: Thanks. We have used “drew” to replaced “reached”.

  1. “Therefore, sand-covered slope was … in wind-water erosion crisscross area”, What does this sentence mean? Please rewrite this sentence clearly.

Response: We rewrote this sentence as follows: “Therefore, sand-covered slopes of different slope gradients were used to simulate the effects of slope and wind erosion. The result will be useful to understand the interaction effect of the slope gradient and wind-blown sand cover on water erosion in the wind–water erosion crisscross region.”

  1. Section 1.1 The soil was composed of 24.8%, … What do you mean by 24.8%? Please give details. 

Response: Sorry,we forgot the word sand, we rewrote this sentence as follows: “The soil was composed of 24.8% sand, 69.4% silt, and 5.8% clay”.

  1. Section 2.1 “and then decreased slowly fluctuations in the relatively stable stage”, What does fluctuations mean in this sentence?

Response: Fluctuations means the rate change, not a fixed value.

  1. Section 2.1 “This phenomenon can be explained by the stored water … .. erosion effect”, Please rewrite this sentence.

Response: We rewrote this sentence as follows: “This phenomenon can be explained by the stored water in the sand layer and soil layer; when it reached its maximum soil water-holding capacity, it was released instantaneously due to gravity and the erosion effect.”

  1. “The process appeared extreme runoff rates on the 20, 25 and 30 slopes”, what do you mean by this sentence? Please rewrite this sentence.

Response: Thanks for your comment. We rewrote this sentence as follows: “The process resulted in extreme runoff rates on the 20°, 25°, and 30° slopes”

  1. “And the total sediment yield of and …  hand a significant from 20”, the authors should clarify had a significant what?

Response: Thanks for your comment. We want to clarify the total sediment yield of different slopes. we rewrote this sentence as follows: “In addition, the total sediment yield of sand-covered slopes showed a significant difference from 20° (P<0.05).”

  1. I cannot understand the meaning of the Figure 8. Please provide more meaningful figure.

Response: Thanks for your comment. We provide more meaningful figure as follows:

R2 means 68% variance of the soil water erosion is explained by the effect of slope and sand cover, with about 75% of it belonging to sand cover and 25% of it belonging to slope.

 Figure 8. Percentage of effects of sand cover and slope on soil water erosion.

Reviewer 2 Report

The set up of the flume box is the most critical part of the experiment. An artificial loess is used and its soil-mechanical  properties are probably not equivalent to the properties of the loess in a natural setting. Therefore the results of the experiment give no reliable data  for the erodability of the loess. The sand cover is less critical but also an analysis of the consistency of the natural sand cover would  be investigated before a conclusion on its stability under rainwater is reached.

the experiment with an artificial loess cannot be used to study the erodability of loess. The experiment should have been done on a undisturbed loess profile.

5. Due to the wrong experimental condition the conclusion are of no value.

Sorry for the authors, I think that they should repeat the experiments
at at undisturbed  loess profile in the country.

Author Response

The setup of the flume box is the most critical part of the experiment. An artificial loess is used, and its soil-mechanical properties are probably not equivalent to the properties of the loess in a natural setting. Therefore, the results of the experiment give no reliable data for the erodibility of the loess. The sand cover is less critical but also an analysis of the consistency of the natural sand cover would be investigated before a conclusion on its stability under rainwater is reached. the experiment with an artificial loess cannot be used to study the erodibility of loess. The experiment should have been done on a undisturbed loess profile.
Due to the wrong experimental condition the conclusion are of no value.

   Response: Thanks for your thought-provoking comment. According to previous studies about mechanism of soil erosion, most studies were used simulated rainfall under the artificial loess slope. Meanwhile under the lab, the conditions of all experiments were to ensure the consistency. We also had considered the review’s suggestion, we will verify the test results in the field. 

Reviewer 3 Report

The manuscript used an experimental model to estimate the soil-water erosion mechanism in loses materials. In general, providing a physical model to analyze the loose soils (with different degrees of saturation) is one of the most important topics in soil and water erosion which is attempted to answer in this manuscript. Referring to the manuscript regardless of the interesting topic of research, there are several modifications that have to be considered. In this regard, the following comments are requested to be addressed by the authors:

1.     The necessity & novelty of the manuscript should be presented and stressed in the “Introduction” section.

2.     Need to provide a cross-section for the slope. Which materials were used, and the condition of the layers (single or multiple)

3.     Spaces should be added between numbers and units. Eg: ' The total runoff volume and the total soil loss increased by 22.8 L and 42.8kg '.

4.     In the experimental design, give the reference of the bulk density of 1.40 g/cm3.

5.     adding the major contribution this paper makes for theory mechanism of soil erosion in " results and discussion " section.

I hope that you will find the comments to be of use to you and am looking forward with interest to receiving your revision.

Author Response

  1. The necessity & novelty of the manuscript should be presented and stressed in the “Introduction” section.

Response: we stressed the novelty in the “Introduction” section. “Therefore, whether the relationship between slope and erosion changes when eroded material changes, and which factor plays a dominant role in the soil water erosion be-tween the slope and wind erosion effect remain unclear. Therefore, sand-covered slopes of different slope gradients were used to simulate the effects of slope and wind erosion.”

  1. Need to provide a cross-section for the slope. Which materials were used, and the condition of the layers (single or multiple).

Response: we add the cross-section in the Figure 1, the sand- cover slope was composed by three layers. The top layer was 2-cm aeolian sand, the second layer was 25-cm loess soil, the bottom layer was 0.5-cm sand.

Figure 1. Experimental devices including water supply pipeline (a), side sprinkler device(b), flume (c), and runoff collection device (d)

  1. Spaces should be added between numbers and units. Eg: ' The total runoff volume and the total soil loss increased by 22.8 L and 42.8kg '.

        Response: Thanks for your comment. We forget the spaces, we also check the whole paper.

  1. In the experimental design, give the reference of the bulk density of 1.40 g/cm3.

Response: Thanks for your comment. We add the reference about the bulk density.

  1. adding the major contribution this paper makes for theory mechanism of soil erosion in " results and discussion " section.

Response: we add one sentence “The result provides a theoretical basis for soil erosion control in this area.” in the abstract. And, also add one sentence “which means the wind effect is more important than the slope factor for soil water erosion in the wind-water erosion Crisscross region.” in the conclusion.

Round 2

Reviewer 2 Report

no further comments
